# Psychological Integrity and Ecological Repair: The Impact on Planetary Public Mental Health (A Narrative Review)

**DOI:** 10.3390/ijerph22101586

**Published:** 2025-10-19

**Authors:** Matthew Jenkins, Sabine Egger

**Affiliations:** 1Department of Psychological Medicine, University of Auckland, Auckland 1023, New Zealand; 2Consultation-Liaison Psychiatry, Waikato District, Health New Zealand, Auckland 3204, New Zealand

**Keywords:** public mental health, planetary health, human rights, ecopsychology, Indigenous frameworks, ecological integrity

## Abstract

Human rights frameworks have historically emphasised physical integrity, yet psychological integrity, the right to mental stability, identity, and emotional safety all remain neglected in health policy and law. This narrative review and commentary argues that psychological integrity is inseparable from ecological integrity, and that contemporary mental health crises are rooted in ruptured human–nature attachments. Drawing on Mother Nature Attachment Theory (MNAT) and supported by emerging empirical evidence, this review traces a trajectory from pre-attachment, through rupture via colonisation, displacement, and ecological collapse, to reconnection through cultural and ecological repair. Gaza exemplifies a contemporary site of deliberate ecological–psychological rupture, where environmental destruction compounds trauma and erodes cultural continuity. In contrast, Indigenous frameworks in Australasia, such as Te Whare Tapa Whā, provide culturally grounded models of reconnection that demonstrate how ecological repair and psychological restoration can proceed together. These contrasting cases illustrate MNAT’s trajectory and emphasise that safeguarding psychological integrity requires embedding ecological security into public health systems. The review concludes that planetary mental health depends on recognising healing of mind and Earth as an indivisible task. Healing mind and Earth must be understood as a single, urgent task within planetary public mental health.

## 1. Introduction

The global health community increasingly recognises that climate change, biodiversity loss, and ecological degradation are among the most pressing determinants of health in the 21st century [1,2,3]. These environmental crises have well-documented physical health impacts, but their psychological consequences are only beginning to be systematically addressed. Recent reviews link climate change and ecological disruption to increased rates of anxiety, depression, trauma, and ecological grief [4,5]. However, while the social determinants of mental health are widely acknowledged, ecological determinants remain underrepresented in psychiatric frameworks. This gap provides the rationale for the present review.

Similarly, the human rights frameworks have long emphasised physical integrity, encompassing the right to bodily security and protection from harm. Yet, psychological integrity, the right to mental stability, emotional safety, and coherent identity remains underdeveloped in both conceptualisation and policy. This review proposes that psychological integrity is inseparable from ecological integrity, and that a failure to recognise and protect this relationship underpins many contemporary mental health crises. In this commentary, building on earlier work on Mother Nature Attachment Theory (MNAT) [6,7], this review positions itself within global public health discourse and offering a framework that connects individual mental health with planetary health, cultural restoration, and environmental justice. Mother Nature Attachment Theory (MNAT) extends attachment theory beyond interpersonal relationships to encompass our relational bonds with the natural world. It proposes that psychological wellbeing emerges from reciprocal participation in ecological systems rather than detachment from them. This perspective is increasingly supported by research in ecological psychology and environmental neuroscience, which demonstrates that nature connectedness enhances wellbeing through affective, cognitive, and physiological pathways [8,9].

The review is written from a region profoundly shaped by colonisation, resource extraction, and the imposition of Western epistemologies upon diverse Indigenous knowledge systems [10,11]. These dynamics, which displaced both human communities and ecosystems, continue to inform the health inequities and ecological degradation experienced today [12]. In our earlier work, it was argued that humans form an attachment to the natural world akin to primary interpersonal attachments: an innate bond which, when ruptured, produces distress and dysfunction at personal and collective levels. This bond can be observed across cultures and throughout human evolution, forming a psychological “secure base” that underpins identity, belonging, and resilience [13,14,15,16,17,18,19,20].

## 2. The Trajectory of Rupture and Repair

MNAT describes a continuum from pre-attachment, in which humans possess an inherent capacity to connect with and derive security from the natural world, through rupture, precipitated by trauma, colonisation, displacement, industrialisation, and ecological collapse, to reconnection, facilitated by cultural repair, ecological justice, and relational restoration [6,7]. MNAT does not rest on an anthropocentric distinction between humans and nature; rather, it assumes their ontological and psychological interdependence. This framework views mental health as emergent from participation within ecological systems rather than dominance over them.

This framing is consistent with empirical research. Recent meta-analyses demonstrate measurable psychological and physiological benefits of exposure to natural environments, including reductions in anxiety, depression, tension, and negative affect, alongside increases in positive mood and vitality [21,22]. Systematic reviews of nature-based interventions confirm their effectiveness in improving mental-health outcomes and promoting wellbeing across diverse populations [23,24]. Umbrella reviews have consolidated evidence that participation in nature-connected activities outside yields consistent improvements in psychological restoration and social connectedness [25]. Conversely, emerging meta-analyses link climate change and ecological disruption to heightened risks of anxiety, depression, and post-traumatic symptoms, particularly in communities experiencing displacement or environmental loss [5,26]. In our Australasian context, rupture is exemplified in the violent dispossession of Māori and Aboriginal peoples from their lands and waters, alongside the commodification and depletion of these ecologies [10,11,12,18]. Such losses severed both material sustenance and cultural narratives, diminishing the intergenerational transmission of ecological identity.

Ecopsychology, environmental psychology, Indigenous scholarship, and environmental neuroscience converge on the finding that human psychological health depends on reciprocal relationships with the environment [14,16,19]. Disruption to these relationships can manifest as depression, anxiety, post-traumatic stress, substance misuse, and even psychotic phenomena, particularly when ecological losses intersect with histories of colonisation and cultural oppression [12,13,14,15,16,18]. This is not merely metaphorical: studies in environmental neuroscience demonstrate measurable cognitive and affective benefits from exposure to natural environments [16,19], while environmental degradation correlates with increased psychological morbidity [15,20,27].

## 3. Gaza as a Contemporary Site of Ecological–Psychological Rupture

The entanglement between people and nature becomes most visible when it is forcibly severed. The ongoing destruction in Gaza provides a contemporary, urgent example. Here, settler–colonial violence and siege conditions produce acute psychological trauma while simultaneously dismantling the ecological and cultural foundations that sustain life [28,29]. Farmland, olive groves, water systems, and coastal ecosystems are destroyed alongside homes and infrastructure, rupturing both immediate survival systems and the deeper attachment bonds between people and their land. This constitutes not only a violation of physical security but also of psychological integrity: eroding the ability to anchor one’s identity, cultural memory, and hope for the future. The mental health impacts are profound and intergenerational, as documented in populations subject to protracted conflict and environmental devastation [13,14,15,16,29].

This dual destruction challenges the artificial separation between humanitarian and environmental responses. In Gaza, ecological collapse is not a collateral consequence but an intentional dimension of warfare [28,29], magnifying psychological harm. The frameworks used to measure and address such harm in global health must therefore account for both the psychological and ecological dimensions of integrity. In contrast, the Australasian context offers insight into how reconnection and cultural repair can begin to restore both ecological and psychological integrity.

## 4. Australasian Positioning in a Global Context

As clinicians and researchers in Aotearoa New Zealand and Australia, this review is situated within settler–colonial states whose own histories of displacement and ecological exploitation remain ongoing [10,11,12,18]. The health systems are shaped by these legacies, often privileging biomedical framings over Indigenous, relational, and ecological perspectives [10,11,18]. Yet this geographical positioning also confers responsibility: Australasia is situated in regions acutely vulnerable to climate change. Comparable ruptures can be observed in climate-affected Pacific nations where sea-level rise threatens place-based identity and cultural continuity, producing forms of ecological grief and existential displacement documented in recent studies [2,20,30].

This proximity to ecological tipping points creates both a moral and professional imperative for Australasian mental health practitioners to engage in planetary health discourse [2,31,32]. Australasia is uniquely placed to integrate Indigenous ecological knowledge, local environmental realities, and global health policy in advocating for psychological integrity as a human right [2,30,32,33]. Our region’s experiences can contribute to an emerging international consensus that climate change, biodiversity loss, and environmental degradation are not only environmental crises but profound determinants of mental health [15,20,27,30].

## 5. Psychological Integrity as a Human Right

Current human rights frameworks inadequately articulate the environmental preconditions for psychological wellbeing [2,30]. The right to health, as codified in instruments such as the International Covenant on Economic, Social and Cultural Rights, implicitly encompasses mental health but seldom recognises the ecological foundations upon which it depends [2,27,30]. The UN Human Rights Council has affirmed the right to a clean, healthy, and sustainable environment [2], but this has yet to be operationalised in mental health policy or clinical frameworks [27,30,31].

This review argues that recognising psychological integrity as a right demands expanding our understanding of environmental determinants. It requires integrating ecological security into public health infrastructure, mental health service design, and disaster response planning [27,30,31,32,33]. This is particularly relevant in contexts like Gaza, where environmental destruction is weaponised [28,29], and in climate-impacted regions across the Pacific, where loss of land and culture threatens both collective identity and individual mental health [15,20,33].

## 6. Reconnection Through Cultural and Ecological Repair

Repairing ruptured attachments to the natural world demands interventions at multiple levels [6,7,10,11,12,13,19,31]. Individually, nature-based therapies, ecological education, and culturally grounded healing practices can foster reconnection [19,31]. At the community level, land-back movements, ecological restoration projects, and Indigenous-led governance create conditions for collective healing [10,11,12,13,18]. Globally, climate justice and biodiversity conservation are integral to sustaining the planetary systems upon which psychological integrity depends [20,33].

Ecotherapy has been integrated into our clinical work in Aotearoa, incorporating therapeutic horticulture and elemental meditation into mental health care [6,7]. These interventions are grounded in MNAT’s recognition that restoring relationship with the natural world can support recovery from trauma, depression, and anxiety [6,7,19,31]. Importantly, such approaches must be culturally tailored: Māori models such as Te Whare Tapa Whā conceptualise health through interconnected dimensions of physical, mental, familial, and spiritual wellbeing, all grounded in land and ancestry [10,12]. Similar frameworks exist in Aboriginal, Pacific, and other Indigenous knowledge systems [10,18], offering culturally specific pathways to reconnection.

## 7. From Local Advocacy to Global Policy

Australasian clinicians can contribute to global health policy by reframing mental health not only as an outcome of social determinants but as an outcome of ecological determinants [20,27,30,32,33]. This reframing challenges public health systems to integrate climate resilience, biodiversity protection, and Indigenous sovereignty into mental health strategies [10,11,12,18]. The WHO’s recent guidance on climate change and mental health [30] offers a platform for this integration. Other policy frameworks, such as the Lancet Countdown on Health and Climate Change, similarly demonstrate how environmental governance and health advocacy can converge to support planetary public mental health [3]. However, further development is needed to ensure that psychological integrity is recognised alongside physical integrity in global health law and practice [2,32].

Such advocacy must also engage with geopolitical realities. In Gaza, in the Pacific, and within our own countries, environmental harm is often inseparable from political structures that perpetuate inequality and displacement [10,11,12,28,29]. Addressing psychological integrity in these contexts requires confronting the political determinants of ecological destruction, including militarisation, extractive economies, and climate inaction [18,20,28].

## 8. Limitations and Research Gaps

This review is narrative in nature rather than systematic and therefore does not claim to provide an exhaustive survey of all relevant literature. The focus has been on displacement as a salient form of ecological rupture, illustrated through selected case studies such as Gaza and Australasia. These examples are intended to demonstrate physical displacement from the land and situate these examples within Mother Nature Attachment Theory (MNAT). It is recognised these are not representative of all global contexts nor does this fully recognise other forms of ecological rupture excluding displacement such as industrialisation and solastalgia [14].

There are also clear gaps in the research base. Empirical studies directly measuring the psychological consequences of ecological rupture remain limited, particularly in displacement and conflict settings. Integration of Indigenous models into global mental health frameworks remains insufficient, despite their potential to guide reconnection. Finally, there is limited evidence on effective interventions at policy and systems level that deliberately target ecological–psychological repair. Addressing these gaps is essential if planetary public mental health is to move beyond conceptual framing toward practical application.

## 9. Conclusions

As summarised in Figure 1, this review has argued that psychological integrity is inseparable from ecological integrity, and that contemporary mental health crises must be understood in relation to ruptured human–nature attachments. Drawing on Mother Nature Attachment Theory (MNAT), it has outlined a conceptual trajectory from pre-attachment, through rupture, to reconnection [6,7]. Gaza has been presented as a contemporary example of deliberate ecological–psychological rupture [28,29], where environmental destruction compounds trauma and erodes cultural continuity [13,14,15,16,17,29]. By contrast, Indigenous frameworks in Aotearoa, such as Te Whare Tapa Whā, illustrate how reconnection may proceed through culturally grounded models of repair and resilience [20,33]. These cases highlight both the destructive impact of ecological collapse and the potential for ecological and indigenous restoration to foster planetary public mental health [10,11,12].

Recognising psychological integrity as a human right requires embedding ecological security into the foundations of health systems [2,32]. Mental health must be reframed as contingent not only on individual and social determinants but also on the stability of ecosystems. This review concludes that healing mind and Earth should be understood as a single, urgent task. The future of planetary public mental health depends on integrating ecological repair with psychological restoration, moving from conceptual recognition toward practical application at policy, community and clinical levels [28,30,31,32,33].

Foundational determinants (ecological integrity, cultural continuity, policy frameworks, community practice, and environmental justice) support the MNAT process, which oscillates between states of sustained attachment, ecological rupture, and Indigenous reconnection. Examples from the Pacific, Gaza, and Australasia illustrate how environmental, geopolitical, and cultural dynamics influence attachment to land. Psychological and ecological integrity, alongside policy and equity, emerge as outcomes of restored planetary health relationships.

## Figures and Tables

**Figure 1 ijerph-22-01586-f001:**
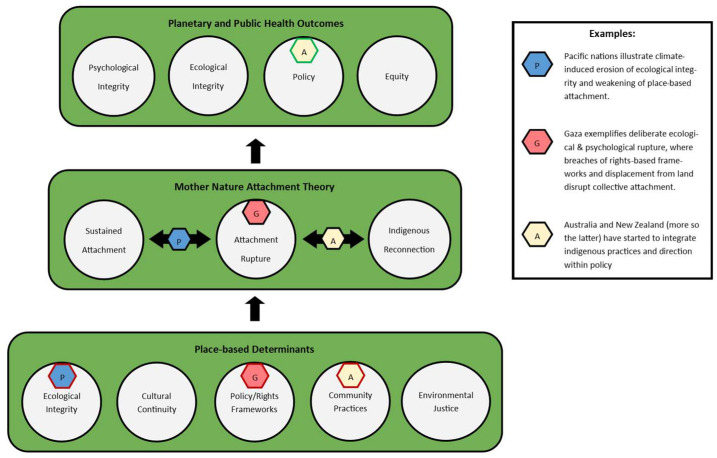
Conceptual integration of Mother Nature Attachment Theory (MNAT) within planetary public mental health.

## Data Availability

No new data were created or analyzed in this study. Data sharing is not applicable to this article.

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
