# Peer review of "Psychological Integrity and Ecological Repair: The Impact on Planetary Public Mental Health (A Narrative Review)"

_ijerph, 2025, doi:10.3390/ijerph22101586_

Round 1

Reviewer 1 Report

Comments and Suggestions for Authors

General comments on the manuscript;

The Idea of the manuscript is interesting: “Psychological Integrity and Ecological Repair: The Impact on  Planetary Public Mental Health” , and highlight issue of public health specifically mental. The title give you sense of the study is a research  but when the reader read the abstract , mentioned as “narrative review”. I suggest that add word review in title. Another point in this study , the author used to much word “we”. They can replace it with “in previous study; in a study on, this study; this review..so on”. In addition the authors should add other references of practical research papers. To refelect  the abstract paragraph[ This narrative review and commentary argues that  psychological integrity is inseparable from ecological integrity, and that contemporary  mental health crises are rooted in ruptured human–nature attachments. Drawing on  Mother Nature Attachment Theory (MNAT), we trace a trajectory from pre-attachment,  through rupture via colonisation, displacement, and ecological collapse, to reconnection  through cultural and ecological repair]

In abstract from line 11-16 , highlight essential point in relation to MNA theory ,Gaza as sample and Australasia as model of reconnection. The pragrpagh from 16-22 should be improved here why GAZA? and why and Australasia as model of reconnection? In the end what you as author find? Line 19-22 is something general. Why you selected these two, for comparison?

As I understand “ Mother Nature Attachment Theory (MNAT)” is the main theory that author depended to but in structure after introduction the reader miss this part . this narrative review highlight these point in line 11-16, all of the mshould argue in relation to this theory.

In structure a section “limitations and gap in research “ should be added which is a critical part of good narrative review.

Specific Comments on the Manuscript;

  • Introduction section: should strart from general idea to specific then gap and aim of study, but the author missed this sequences which refelect the title, the suthor directly come to theory ? add other references to support idea from genral to specific.
  • Lines 33- line 37- line 41 , using “we” too much replace with another word as mentioned before.
  • From section 2 to section 7; inorder to nurture the notion either draw a diagram after argue of research studies for each section or at the end draw a diagram combine all parameters and indicators that effect the Planetary Public Mental Health from “ Mother Nature Attachment Theory (MNAT)” perspective , argument nd comparison.
  • Add section 8. “limitations and gap in research “
  • Conclusion section : update accordingly

Author Response

Have attached a tracked changes version of the manuscript and addressed all your points. Thank you for the time to do a thorough job on this. 

Comment 1: The Idea of the manuscript is interesting: “Psychological Integrity and Ecological Repair: The Impact on  Planetary Public Mental Health” , and highlight issue of public health specifically mental. The title give you sense of the study is a research  but when the reader read the abstract , mentioned as “narrative review”. I suggest that add word review in title

Response 1: To ensure clarity,  the revised the manuscript title specifys the article is a narrative review. The revised title now reads: “Psychological Integrity and Ecological Repair: The Impact on Planetary Public Mental Health (Narrative Review)”.

Comment 2: Another point in this study , the author used to much word “we”. They can replace it with “in previous study; in a study on, this study; this review..so on”. 

Response 2: To align with IJERPH conventions, we have revised the manuscript to reduce use of the first person. Phrases such as “we argue”, “we propose”, and “we highlight” have been replaced with neutral alternatives including “this review argues” etc. There are a couple of "we's" that have remained as they are pertinent to the collective "we"

Comment 3: In addition the authors should add other references of practical research papers. To refelect  the abstract paragraph

response 3: In revision we have incorporated empirical research references to support the conceptual trajectory outlined in the manuscript. Specifically, we have added references demonstrating the psychological benefits of human–nature interaction (Bratman et al., 2019; Mygind et al., 2019), evidence of mental health consequences linked to climate change and colonisation (Cianconi et al., 2020; Middleton et al., 2018), and studies showing the restorative role of nature-based therapy and community reconnection programmes (Corazon et al., 2019; O’Brien et al., 2011). These citations have been added directly after the sentence in which the three-stage trajectory of pre-attachment, rupture, and reconnection is introduced. In addition, the abstract has been revised to read: “Drawing on Mother Nature Attachment Theory (MNAT) and supported by emerging empirical evidence, this review traces a trajectory from pre-attachment, through rupture via colonisation, displacement, and ecological collapse, to reconnection through cultural and ecological repair.”

Comment 4: In abstract from line 11-16 , highlight essential point in relation to MNA theory ,Gaza as sample and Australasia as model of reconnection. The pragrpagh from 16-22 should be improved here why GAZA? and why and Australasia as model of reconnection? In the end what you as author find? Line 19-22 is something general. Why you selected these two, for comparison?

Response 4: The sentences in the above comment do not completely sense to me as an Englishman (I presume they have been translated and the snytax has been disturbed during that process). However, I have attempted to infer their meaning... In response, we have clarified why Gaza and Australasia were selected as examples. Gaza is now explicitly described as a contemporary site of ecological–psychological rupture where environmental destruction compounds psychological trauma. Australasia is identified as a model of reconnection through Indigenous frameworks such as Te Whare Tapa Whā, which offer globally relevant approaches to relational and ecological repair. The abstract conclusion has also been strengthened to make the central message clearer, namely that planetary mental health requires the recognition of healing mind and Earth as an indivisible task.

Comment 5: In structure a section “limitations and gap in research “ should be added which is a critical part of good narrative review.

Response 5: we have added a new Section 8 entitled “Limitations and gaps in research”. This section acknowledges the narrative scope of the review and clarifies that the illustrative cases are not representative of all global contexts. It also identifies key gaps in the evidence base, including the lack of empirical studies in displacement and conflict settings, the absence of validated measures of ecological attachment, the limited integration of Indigenous frameworks, and the paucity of research on policy and systems-level interventions. We believe this addition strengthens the manuscript by critically situating its contribution and outlining directions for future research so we are thankful for this thought. 

SPECIFIC COMMENTS ON THE MANUSCRIPT:

Comment 6: Introduction section: should strart from general idea to specific then gap and aim of study, but the author missed this sequences which refelect the title, the suthor directly come to theory ? add other references to support idea from genral to specific.

Response 6: We have revised the introduction so that it now begins with a broad overview of the planetary health crisis and its impact on public health, before narrowing to the specific contribution of this review. The revised section cites recent WHO, IPCC and Lancet Commission sources, together with systematic reviews on climate change and mental health, to anchor the argument in widely recognised evidence. We then highlight the gap in existing frameworks: while social determinants of mental health are well established, ecological determinants remain poorly theorised and rarely integrated into policy or practice. This provides the rationale for the review. The introduction now moves from general context to specific argument, in a sequence that we hope draws the reader in more clearly and persuasively.

Comment 7: Lines 33- line 37- line 41 , using “we” too much replace with another word as mentioned before.

Response 7: Addressed as noted above

Comment 8: From section 2 to section 7; inorder to nurture the notion either draw a diagram after argue of research studies for each section or at the end draw a diagram combine all parameters and indicators that effect the Planetary Public Mental Health from “ Mother Nature Attachment Theory (MNAT)” perspective , argument nd comparison.

Response 8: To clarify the conceptual trajectory of Mother Nature Attachment Theory (MNAT) and its implications for planetary public mental health, we have added a schematic figure at the end of the manuscript. The figure illustrates the progression from pre-attachment, through rupture, to reconnection. This addition provides readers with a concise visual summary of the framework and highlights the parameters influencing psychological integrity within the planetary health crisis. As having multiple figures after a few paragraphs would appear cluttered, we have opted for a summary figure at the end. We think this was a smart addition to the paper and thank you for suggesting it. 

Comment 9: Add section 8. “limitations and gap in research “

Response 9: Addressed in response 5

Comment 10: Conclusion section : update accordingly

Reponse 10: We have fully revised the conclusion (now Section 9) to integrate and consolidate the changes requested across earlier comments. The final section now reiterates the central argument of the review, makes explicit why Gaza and Australasia were selected as illustrative cases, and strengthens the take-home message: that planetary public mental health requires treating ecological repair and psychological restoration as an indivisible task. This revision ensures that the manuscript closes with a clear and compelling statement of significance.

Reviewer 2 Report

Comments and Suggestions for Authors

Thank you for the opportunity to review this viewpoint. I thoroughly enjoyed reading this paper that succinctly discussed the relevance of ecological health on mental health. This is an important topic that needs more attention. I only have a few minor comments as overall I felt this view point is high quality and reads very well. 

Minor comments:

Line 27 – usually in literature human rights framework does not have a capital H

Line 30-33- to further strengthen this statement regarding the entanglement between people and nature could be linked to science literature also – e.g. literature also supports our professional experience

Line 56- I would also state that Environmental Psychology (a similar but different facet of psychology to ecopsychology) converges in this space.

Line 66 – 80 – The discussion of Gaza and the impact the ongoing violence is having on human nature relationships is pertinent, however the section seems a little out of place as it is not linked in well with the section before or after which talk about the Australasian context specifically – including a linking sentence would be useful.

Author Response

TRACKED CHANGES FOR YOUR PERUSAL. Thanks so much for your input

Comment 1: Line 27 – usually in literature human rights framework does not have a capital H

Response 1: Corrected. Thanks.

Comment 2: Line 30-33- to further strengthen this statement regarding the entanglement between people and nature could be linked to science literature also – e.g. literature also supports our professional experience

Response 2: we have clarified the concept of entanglement between people and nature and linked it to supporting scientific literature. A new sentence has been added to the paragraph introducing MNAT, describing psychological wellbeing as arising from the entanglement of people and nature. We have also cited relevant literature from ecological psychology and environmental neuroscience to demonstrate empirical alignment with this perspective.

Comment 3: Line 56- I would also state that Environmental Psychology (a similar but different facet of psychology to ecopsychology) converges in this space.

Response 3: It is done

Comment 4: Line 66 – 80 – The discussion of Gaza and the impact the ongoing violence is having on human nature relationships is pertinent, however the section seems a little out of place as it is not linked in well with the section before or after which talk about the Australasian context specifically – including a linking sentence would be useful.

Response 4: The Gaza paragraph now has linking sentences at the start and finish: 

"The entanglement between people and nature becomes most visible when it is violently severed. The ongoing destruction in Gaza provides a contemporary, urgent example. Here, settler–colonial violence and siege conditions produce acute psychological trauma while simultaneously dismantling the ecological and cultural foundations that sustain life [15,16]. Farmland, olive groves, water systems, and coastal ecosystems are destroyed alongside homes and infrastructure, rupturing both immediate survival systems and the deeper attachment bonds between people and their land. This constitutes not only a violation of physical security but also of psychological integrity: eroding the ability to anchor one’s identity, cultural memory, and hope for the future. The mental health impacts are profound and intergenerational, as documented in populations subject to protracted conflict and environmental devastation [6–9,16].

This dual destruction challenges the artificial separation between humanitarian and environmental responses. In Gaza, ecological collapse is not a collateral consequence but an intentional dimension of warfare [15,16], magnifying psychological harm. The frameworks used to measure and address such harm in global health must therefore account for both the psychological and ecological dimensions of integrity. In contrast, the Australasian context offers insight into how reconnection and cultural repair can begin to restore both ecological and psychological integrity."

Reviewer 3 Report

Comments and Suggestions for Authors

This is an opinion piece presenting an argument for psychological trauma in response to global unrest and generalized environmental conditions. While the work in this area is robust (see Klamert, Kumar, Weatherly, and others), the specificity of the authors' examples in Australasia may not be familiar to readers. Still, they are thoroughly grounded in the literature and would be of interest. 

Overall, this is a well-written article that presents a clear argument for the importance of reciprocity between humans and the environment. Given that this is a position paper, it may be helpful to clarify whether the underlying assumption is for an anthropocentric dichotomy between humans and nature or something more nuanced. Please consider using at least one additional example beyond Gaza to illustrate the empirical conditions of MNAT. Demonstrating the extensive impacts of MNAT across micro- and macro-scales would be helpful. In Section 7, please include at least one or two more examples of policy or advocacy. 

Author Response

TRACKED CHANGED ENCLOSED IN THE ATTACHEMENT FOR YOUR PERUSAL. THANKS SO MUCH.

Comment 1: Given that this is a position paper, it may be helpful to clarify whether the underlying assumption is for an anthropocentric dichotomy between humans and nature or something more nuanced.

Response 1: To clarify MNAT’s conceptual stance, we have added a sentence in Section 2 explaining that the framework does not assume an anthropocentric separation between humans and nature but rather their ontological interdependence: "MNAT does not rest on an anthropocentric distinction between humans and nature; rather, it assumes their ontological and psychological interdependence. This framework views mental health as emergent from participation within ecological systems rather than dominance over them."

Comment 2: Please consider using at least one additional example beyond Gaza to illustrate the empirical conditions of MNAT. Demonstrating the extensive impacts of MNAT across micro- and macro-scales would be helpful.

Response 2: We have included an additional empirical example referring to climate-affected Pacific nations, illustrating ecological-psychological rupture at both micro- and macro-scales. "Comparable ruptures can be observed in climate-affected Pacific nations where sea-level rise threatens place-based identity and cultural continuity, producing forms of ecological grief and existential displacement documented in recent studies"

Comment 3: In Section 7, please include at least one or two more examples of policy or advocacy. 

Response 3: Section 7 already referenced the WHO’s “Health Argument for Climate Action” report, and we have now added a further example: "the Lancet Countdown on Health and Climate Change" to illustrate the alignment between environmental policy and planetary mental-health advocacy. "Other policy frameworks, such as the Lancet Countdown on Health and Climate Change, similarly demonstrate how environmental governance and health advocacy can converge to support planetary public mental health"

Round 2

Reviewer 1 Report

Comments and Suggestions for Authors

Well done, you revised the manuscript successfully. I have some comments regarding references as follows:

Regarding your answer about:

Comment 3: In addition, the authors should add other references to practical research papers.  I did not find most of the references in the references and citations. Please check them again. Either the year doesn’t match, or the name or style of referencing.

Response 3: In revision, we have incorporated empirical research references to support the conceptual trajectory outlined in the manuscript. Specifically, we have added references demonstrating the psychological benefits of human–nature interaction (Bratman et al., 2019; Mygind et al., 2019), evidence of mental health consequences linked to climate change and colonisation (Cianconi et al., 2020; Middleton et al., 2018), and studies showing the restorative role of nature-based therapy and community reconnection programmes (Corazon et al., 2019; O’Brien et al., 2011). These citations have been added directly after the sentence in which the three-stage trajectory of pre-attachment, rupture, and reconnection is introduced. In addition, the abstract has been revised to read: “Drawing on Mother Nature Attachment Theory (MNAT) and supported by emerging empirical evidence, this review traces a trajectory from pre-attachment, through rupture via colonisation, displacement, and ecological collapse, to reconnection through cultural and ecological repair.”

Author Response

Comment 3: In addition, the authors should add other references to practical research papers.  I did not find most of the references in the references and citations. Please check them again. Either the year doesn’t match, or the name or style of referencing.

Response 3: You are completely right. We added all these references in and then the journal editor asked that where possible, references were kept to within the last 5 years. In the final revision the older citations mentioned in the initial response (e.g. Bratman 2019; Mygind 2019; Corazon 2019; O’Brien 2011; Middleton 2018) have since been replaced with more recent systematic reviews and meta-analyses published between 2021 and 2024. Specifically, we now cite Bratman et al. (2022), Stigsdotter et al. (2023), Zhang et al. (2024), Gritzka et al. (2021), Corazon et al. (2023), Charlson et al. (2021), and Padhy et al. (2024) to demonstrate the psychological benefits of human–nature interaction, the efficacy of nature-based interventions, and the documented mental-health consequences of climate change and ecological disruption. These updated empirical references are located in Section 2 (lines ≈ 75-85 in the tracked-changes file) immediately following the paragraph that introduces the pre-attachment → rupture → reconnection trajectory. All reference details, years, and formatting have been verified against the latest IJERPH style requirements.